# Influence of the Urban Intersection Reconstruction on the Reduction of Road Traffic Noise Pollution

**Dusan Jandacka ***, **Martin Decky, Katarina Hodasova, Peter Pisca and Dusan Briliak**

Department of Highway and Environmental Engineering, Faculty of Civil Engineering, University of Zilina, Univerzitná 8215/1, 010 26 Zilina, Slovakia
* Correspondence: dusan.jandacka@uniza.sk

**Abstract:** The authors present the unsolved issue of the contextual design of urban intersections (UI) from the point of view of traffic noise emissions around residential buildings in the Slovak context. Noise barriers are very rarely used in urban areas, due to such reasons as lack of space for their installation, traffic safety (view at intersections), architectural factors, as well as the fact that they represent a barrier for pedestrians and cyclists. The use of low-noise asphalt pavements is limited in urban areas primarily due to the high cost of production and maintenance of these covers, their limited durability in a colder climate, and lower efficiency compared to the roads outside urban areas. From this aspect of urban intersection design, the authors focused on the identification of individual factors associated with the significant reduction (2–8 dB) of traffic noise induced by the reconstruction of non-signalized urban intersections into roundabouts. The authors draw attention to the fact that both new surfaces of intersection branches and the change in traffic processes contribute to the aforementioned reduction. This finding was obtained by comparing direct measurements of noise levels and the results of their predicted values using validated 3D models in the CadnaA program. Noise emissions were measured by a noise analyzer (NOR-1210), and traffic noise emissions were predicted by the "Nouvelle Méthode de Prévision du Bruit" methodology (NMPB-1996). Based on the analysis of the measured and predicted traffic noise levels, the authors objectivized the share of reduction in traffic processes in the 2–3 dB range. The authors objectified the share of traffic noise reduction due to the change in traffic processes as being in the 2.2–3.3 dB range. The presented research results could contribute to a quantifiable reduction in the noise load in the external environment of residential buildings using the contextual design of intersections.

**Keywords:** contextual design; intersection; roundabout; equivalent noise level; residential buildings; pavement; road reconstruction

## 1. Introduction

In general, the external environment around residential buildings is perceived by several modalities, including sight, sound, and touch [1]. In general, residents of residential areas consider anthropogenic noise to be the most disturbing emission limiting the quality of the environment [2–6]. The ongoing spread of urban areas, highways, and airports throughout the world makes anthropogenic noise almost omnipresent [2]. Traffic noise is perceived by the public as one of the most disturbing types of anthropogenic noise, and its negative effects on public health, including annoyance, a reduction in mental health, hypertension, and an increased risk of myocardial infarction, have been demonstrated by many authors [3–7]. In EU countries, the issue of noise has become a professional and political topic in the past two decades. In recent years, several European research projects have addressed the issue of road traffic noise, including SILVIA (Sustainable Road Surfaces for Traffic Noise Control), INQUEST (Information Network Quiet European Road Surface Technology), and SPENS (Sustainable Pavements for European New Member States) [8–11]. In the aforementioned projects, the main focuses were the measurement of the influence of road surfaces on traffic noise and the effect of low-noise pavements [12–15].

Optimizing the surface texture at the macro scale was found to be important for reducing tire/road noise. Regarding pavement types, porous asphalt concrete and its variants have the most reliable low-noise properties while also having some drawbacks in terms of durability and maintenance [13]. The noise level at thin-surface layers tends to be 3 dB lower for cars and 1 dB lower for heavy vehicles than for average asphalt concrete/stone mastic asphalt. Porous pavements yield average noise reductions in the order of 3–4 dB. The potential noise reduction, as reflected by the lowest noise level measured for a family of pavements is in the order of 5–10 dB for thin-layer surfaces and porous asphalt [14]. After 3 years of service, the sound level ($L_{Amax}$) of the low-noise surfaces of very thin asphalt concrete and porous asphalt concrete was shown to increase by 5.7 to 6.5 dB to a level similar to that of standard asphalt concrete [15]. Based on a literature review, the authors feel that the contextual design (CD) of new, or the rehabilitation of existing, urban crossroads (UC) is not used enough to minimize the noise load of the surroundings.

Contextual design (CD) is a user-centered design process that was developed by Hugh Beyer and Karen Holtzblatt. It is a step-by-step process in which field data is collected and used to design any sort of product that includes a technical component [16,17]. CD has primarily been used for the design of computer information systems, including hardware and software. Parts of contextual design have been adapted for use as a usability evaluation method and for contextual application design [18,19]. The use of CD has been increasingly promoted in the fields of urban planning, architecture [20–22], and civil engineering [23,24]. This paper uses Slovak conditions to present a new approach to the design of UC, which integrates acoustic, psychoacoustic, architectural, environmental, and economic aspects, providing a holistic approach [25–27] to the design of these engineering structures (CD of UC), similar to what has been used in the crossroads contextual design for Polish [28–31] and Czech [32] conditions, as well as by other authors [33–38]. The authors of a previous study [27] presented the holistic approach as a sustainable way to prepare, construct, and manage integrated transport infrastructure with a particular focus on pavement in the middle Europe area. The situation in areas exceeding traffic noise limits in Slovakia is very similar to that in Poland. In another article [30], it is stated that, in Poland, the acceptable value according to the law for the night-time sub-interval is exceeded by about 11 dB. From the aspect of noise emissions, non-signalized roundabouts are believed to have a significantly lower noise load than non-signalized intersections. However, the situation is significantly different in the case of light-controlled roundabouts. According to [31], the noise levels at a distance of 60 m from the central point of intersections with comparable traffic volumes are higher for signalized roundabouts by 3.3–6.7 dB in relation to non-signalized roundabouts. According to researchers [33], one way to reduce road noise in residential areas is to regulate road traffic at intersections. A comparative analysis between signalized intersections and roundabouts under the same road traffic flow revealed a reduction in noise pollution of 1–2 dB for roundabouts [34].

Another study [35] showed that the improvement of traffic fluidity (for example, by roundabouts) can reduce noise by 2–4 dB [36]. The reconstruction of a signalized intersection to a roundabout makes a reduction of 1 dB possible. The noise reduction from cars approaching the roundabout at a lower speed is 5 to 10 dB; however, this is compensated by the noise increase of cars accelerating when leaving the roundabout, producing an increase in noise of 3 to 8 dB [35].

A study [37] showed that roundabouts lead to a reduction in noise of between 3–4 dB compared with that of standard intersections. The impact of a roundabout on the noise level and its applicability as a traffic calming device and a noise abatement measure should be investigated in the early design stage by modeling noise levels [38]. Recently, many authors have described various Context Sensitive Multimodal Assessment (COSMA) methods for the purpose of improving road design. The context is defined in terms of a range of land use, socioeconomic, environmental, transportation, and psychoacoustic factors that are presented spatially [39–41]. Another paper [32] addressed the COSMA design of an MCA

(Multi-criteria assessment) methodology for at-grade intersections in both urban and rural areas of the Czech Republic. In urban areas, such methods should constitute the deciding element related to the design of the network.

From the broad issue of MCA contextual urban road design, the authors set themselves the goal of refining the input of noise prediction in the vicinity of non-signalized intersections. These can be used by road designers to develop proposals for road infrastructure design to minimize negative impacts on the ecosystems of residential buildings, allow road authorities to improve the environmental safety of residents around the roads, and facilitate the management of green residential buildings [42,43]. The authors of this article believe that, in the current period, the immediate surroundings of residential buildings should be perceived as an integral part of human habitats, as one of the ecosystems of the inclusive urban environment. The authors assume full convergence of the presented issue with Transforming our world: The 2030 agenda for sustainable development [44], especially regarding Goal 11 (Make cities and human settlements inclusive, safe, resilient and sustainable).

To fulfill the set goal, comparisons of measured and predicted traffic noise emissions in the vicinity of non-signalized intersections before and after reconstruction into roundabouts were made. At the same time, relevant characteristics of the traffic streams of all intersection arms (speed and proportion of passenger and truck vehicles) influencing the propagation of noise in the outdoor environment of residential buildings were investigated. In all considered cases, the measured values of noise emissions were evaluated (measurements realized in Michalovce city and Žilina city), and in the case of the reconstruction of a three-arm intersection into a roundabout in the city of Žilina, a model was created in the CadnaA program. The "Nouvelle Méthode de Prévision du Bruit" methodology (NMPB-1996) was used to create a noise propagation model around intersections [45,46]. In all evaluated cases of the reconstruction of intersections into roundabouts, a decrease in the noise level of the surroundings was observed (results from measurements and modeling). This study reveals the possible impact of the contextual design of non-signalized intersections on the environment of residential buildings from the point of view of the propagation of noise emissions from road traffic.

## 2. Materials and Methods

### 2.1. Basic Principles of Traffic Noise Emissions

To objectify the negative impact of road transport on public health, the operator of the transport infrastructure is obliged to ensure that the traffic noise is within the admissible limits according to Act no. 549/2007 Coll. [47]. We present more details about this issue in [48–50]. In this contribution, we focus primarily on methods to ensure admissible noise limits are met in the outdoor environments of residential buildings. We use the Slovakian context for the assessment of road traffic noise. The decisive characteristic is the equivalent level $L_{Aeq}$ of sound (noise). Equation (2) is based on the basic Equation (1) for determining the sound pressure level.

$$L = 10 \cdot \log(p/p_o)^2 \tag{1}$$

where:

$p$ is the sound pressure [Pa],
$p_o$ is the reference sound pressure, $p_o = 2 \times 0^{-5}$ Pa.

The noise level from road traffic $L_{Aeq}$ (dB) is at a continuous sound pressure level $p_A(t)$ indicated by traffic flow, which is corrected by the frequency weighting function A (Figure 1) and calculated according to Equation (2).

$$L_{Aeq} = 10 \cdot \log \frac{1}{T} \int_{t_1}^{t_2} \left[ \frac{p_A(t)}{p_o} \right]^2 \cdot dt \tag{2}$$

where:

$p_A(t)$ is the time function of pressure sound weighted by frequency weighting function A (Figure 1),

$T$ is the integration interval, $T = t_2 - t_1$ [s].

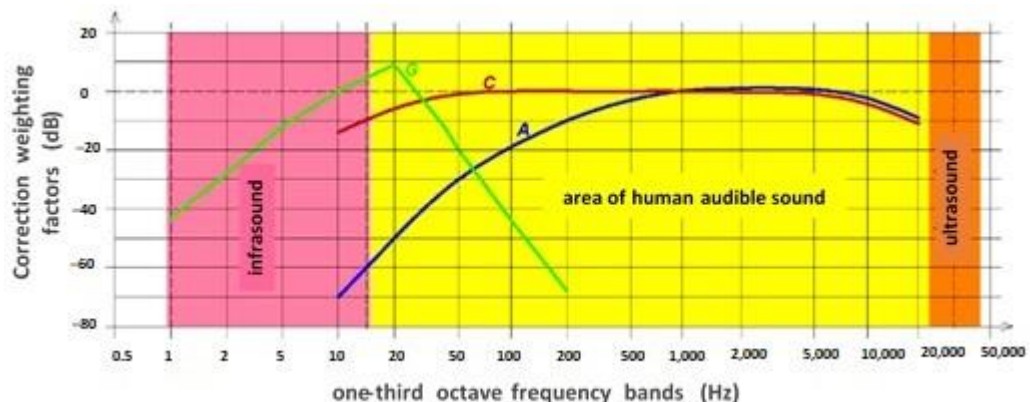

**Figure 1.** Frequency weighting characteristics for the assessment of human exposure to noise [49].

This article presents a frequency analysis of traffic noise in the one-third band, for which the upper limit frequency $f_u$ (Hz) is equal to $2^{1/3}$ times the lower limit frequency $f_l$ (Hz), which means that Equation (3) applies.

$$f_u = 2^{1/3} \cdot f_l \tag{3}$$

### 2.2. Admissible Road Traffic Noise Levels in the Slovak Republic

Although this article only mentions the traffic noise effects on the outdoor environment of residential buildings, the authors consider the outdoor environment to be an integral part of the buildings. The public generally perceives traffic noise as having the most disturbing influence on the quality of the external environment, and it can be concluded with a high degree of probability that noise from road traffic is a limiting factor in the quality of housing. In strategic noise mapping and noise studies, traffic noise is evaluated separately for reference time intervals: daytime is from 6 a.m. until 6 p.m., evening is from 6 p.m. until 10 p.m., and night-time is from 10 p.m. until 6 a.m. The decisive codified provisions regarding the method of determining the admissible limits of traffic noise in the external environment (Table 1) and the indoor environments of buildings are presented in [47]. This Decree of the Ministry of Health of the Slovak Republic [47] states that the protection of public health against traffic noise is ensured if the quantities of noise for the day, evening, and night are not higher than the noise limits presented in Table 1.

**Table 1.** The traffic noise admissible (a) values for selected outdoor territory categories around residential buildings [47].

| Territory Category | Description of the Outdoor Environment | Time Interval | Admissible Values (dB) | | | |
|---|---|---|---|---|---|---|
| | | | Traffic Noise | | | |
| | | | Road and Water Transport $L_{Aeq,a}$ | Railways $L_{Aeq,a}$ | Air Transport | |
| | | | | | $L_{Aeq,a}$ | $L_{ASmax,a}$ |
| II. | Areas in front of windows of residential rooms; protected rooms, such as school buildings, medical facilities, and other protected buildings; and outdoor territories in residential and recreational areas | **day** **evening** **night** | **50** **50** **45** | 50 50 45 | 55 55 45 | - - 65 |
| III. | Areas in category II near motorways and roads of classes I and II, local public transport roads, railways, airports, and city centers | **day** **evening** **night** | **60** **60** **50** | 60 60 55 | 60 60 50 | - - 75 |

Admissible values for the noise level in the indoor environments of buildings [47] are defined in the following way. The determining factor for indoor building environments is the maximal *A* sound level $L_{Amax}$ or the equivalent *A* sound level $L_{Aeq}$.

Admissible values for the noise level in indoor environments of residential buildings from external sources according to [47] are 40 dB for the day and evening, and 30 dB for the night.

### 2.3. Study Area Used for the Reconstruction of Non-Signalized Intersections into Roundabouts from the Aspect of Noise Pollution

As part of this contribution, we focused on four intersections that were transformed from non-signalized intersections into roundabouts in Michalovce City and Žilina City from the point of view of the change in road-traffic-related noise emissions.

Photos of the intersections before and after reconstruction from non-signalized intersections into roundabouts (R1-R3) are presented in Figure 2. In the outdoor environment of residential buildings in Michalovce in the period 2007–2012 (day of measurements are specified in Section 3.1), the following intersections were reconstructed:

- Štefánikova–Saleziánov–Martina Rázusa streets (R1);
- Štefánikova–Okružná–Jána Švermu streets (R2);
- Jána Hollého–Moskovská–Okružná—Lastomírska streets (R3).

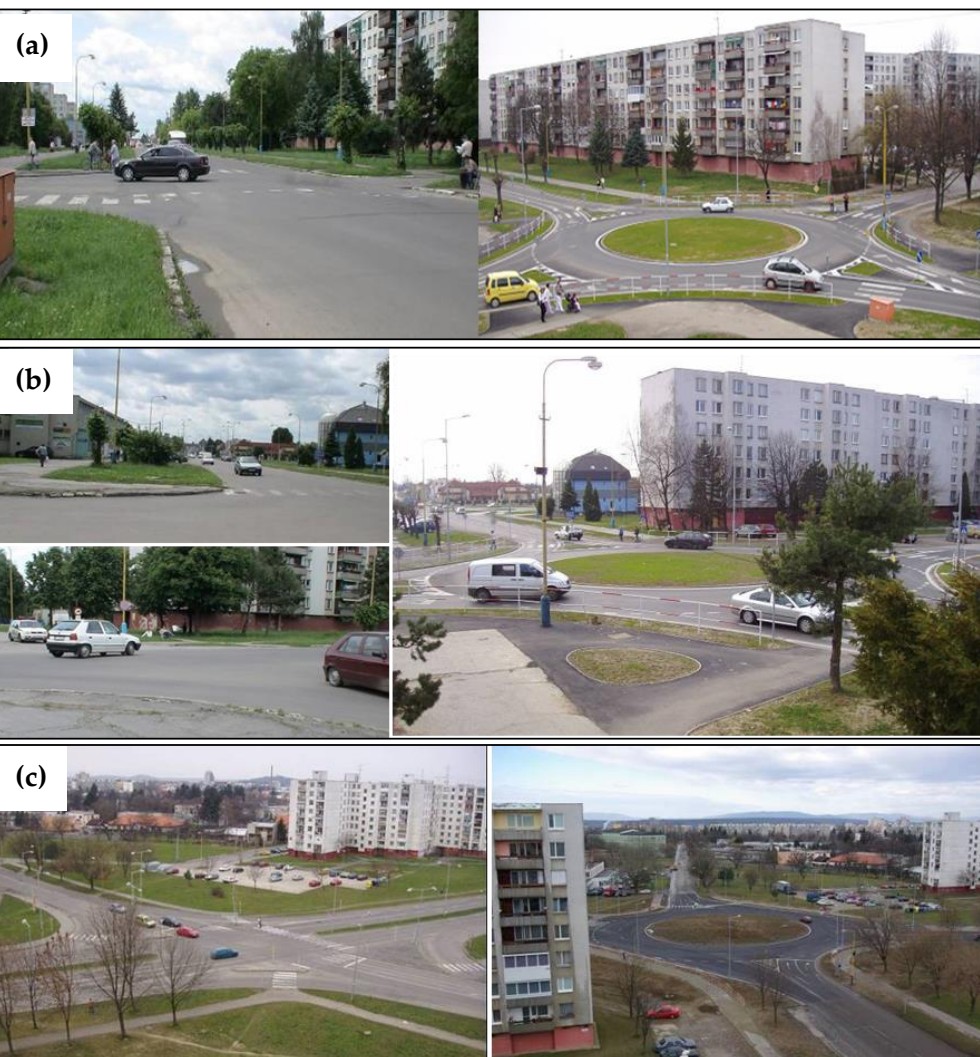

**Figure 2.** Photos of the intersections (**a**) R1, (**b**) R2 and (**c**) R3 before (**left**) and after (**right**) reconstruction to roundabouts in Michalovce City.

All intersections in the city of Michalovce were non-signalized four-arm intersections before reconstruction.

At the mentioned intersections, a traffic survey was carried out at the same time as the noise load monitoring. The results (before and after reconstruction) of the traffic volume (veh/h) (vehicles entering the intersection) and the truck proportion *p* (%) are presented in the Table 2. Differences of up to 8% were found between the presented traffic volumes, which created a successful basis for correct comparison of noise load before and after the reconstruction of intersections.

**Table 2.** Hourly traffic volume and truck proportion at the intersections detected during noise load measurements in Michalovce City.

| Intersection | Before | | After | |
|---|---|---|---|---|
| | Traffic Volume (veh/h) | *p* (%) | Traffic Volume (veh/h) | *p* (%) |
| R1 | 1091 | 2.5 | 1006 | 2.6 |
| R2 | 1235 | 2.8 | 1156 | 2.6 |
| R3 | 885 | 3.8 | 903 | 3.6 |

The basic characteristics of two roundabouts in Michalovce were diameters of *D* = 31 m, single lane widths of 5.5 m, and wearing course from asphalt concrete AC11 wear (Figure 2a,b). The third roundabout had a diameter of 62 m and two lanes on the circuit with a total width of 12 m and wearing course from asphalt concrete AC11 wear (Figure 2c).

Another study of the impact on the noise load of the reconstruction of a three-arm intersection into a roundabout with a diameter of 36 m and a single lane width of 6 m in the vicinity of residential buildings took place in the city of Zilina (Figure 3). The reconstruction was carried out in 2016. The traffic noise emission measurements at the three-arm intersection were carried out in 2016 (T_16), and the measurements were carried out at two measuring stations: MS1 (29 June 2016, 8:45–9:45 a.m.) and MS2 (29 June 2016, 10:00–11:00 a.m.). The original three-arm intersection was located at the point where the industrial and recreational zones of the city of Zilina intersect with a built-up area of companies and recreation facilities and services.

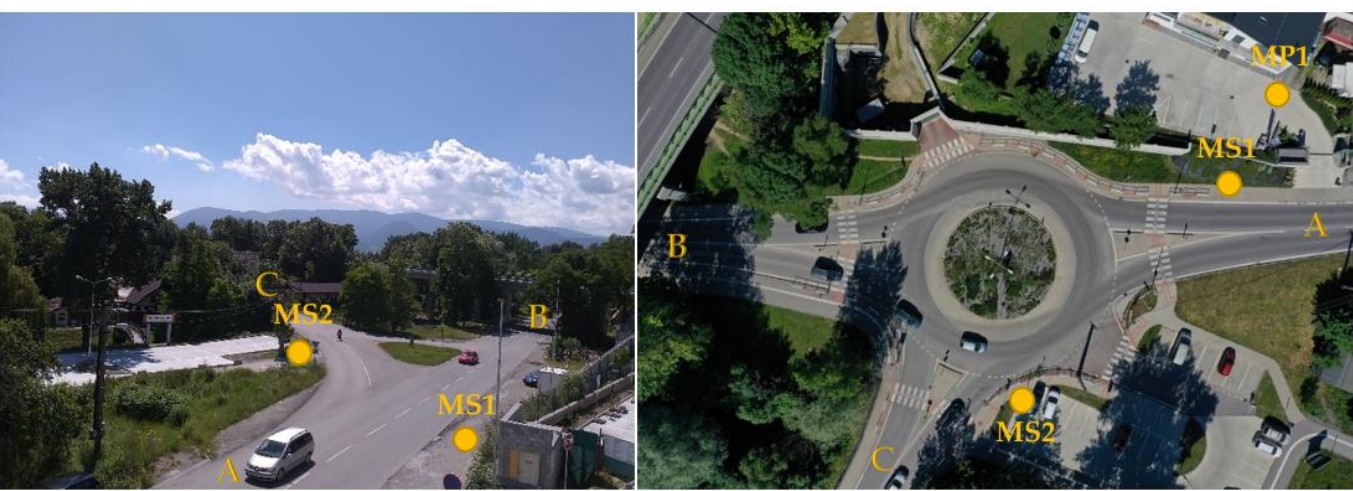

**Figure 3.** Photos of the three-arm intersection before (**left**) and after (**right**) reconstruction into a roundabout and marked measuring stations and intersection arms in Žilina City.

Traffic noise emission measurements at the new roundabout were carried out in 2017 (R_17). The measurements were carried out at the same measuring stations as used for the original three-armed junction: MS1 (28 June 2017, 8:45–9:45 a.m.) and MS2 (28 June 2017, 10:00–11:00 a.m.) (Figure 3). The main purpose of the presented reconstruction was to construct a safe, capable, and environmentally satisfactory crossroad as a traffic hub for

vehicles as well as for pedestrians and cyclists. The roundabout was built with a connection to existing residential and recreational territories and industrial areas of Zilina, significantly contributing to an increase in traffic safety in the location of interest.

Another supplementary measurement was carried out at the roundabout at measuring stations MS1 (15 June 2022, 9:00–10:00 a.m.) and MS2 (15 June 2022, 10:15–11:15 a.m.) in 2022 after five years of use (R_22). During the measurements of the noise levels, the characteristics of the traffic flow (traffic volume, speed, and composition) were also monitored using radar counting devices (for more detailed information, see Section 2.5).

### 2.4. Objectification of the Impact of the Reconstruction of Intersections into Roundabouts on the Noise Load in the Vicinity of Residential Buildings

The presented traffic noise measurements for $L_{Aeq}$ were attained with a sound analyzer (NOR-121, accuracy Class 0, Figure 4) with verification and calibration of the sound level meter, a measuring microphone (by the NORSONIC N-121 calibrator, accuracy Class 1), and one-third-octave filters. Traffic noise monitoring was carried out in accordance with the provisions of the corresponding standards [51,52] valid at the time of measurement. During the measurements, in addition to the already mentioned calibrations, the following observations were made: the microphone met the required scale needs and was primarily placed on the tripod and covered for protection from dust and wind (Figure 4). During measurements, people were prevented from getting close to the microphone, and measurements were carried out under wind speed conditions of up to 5 m.s$^{-1}$, no rain or snowfall, and in temperatures of over 5 °C.

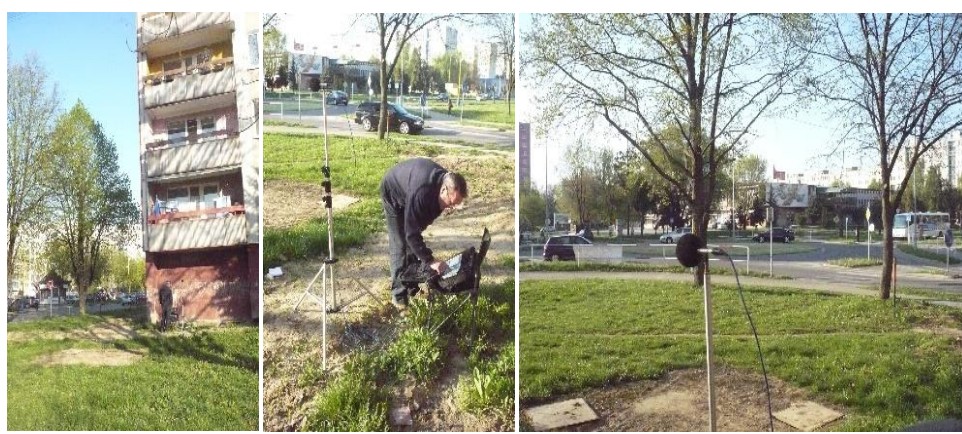

**Figure 4.** Photos of the measuring set sound analyzer (NOR-121, accuracy Class 0) when measuring roundabout noise emissions at R2.

### 2.5. Modelling Noise

The CadnaA program was used to model the noise emissions from road traffic in the vicinity of the intersection in Žilina City. Three scenarios of real external environments around intersections in the built-up area were modeled: (a) a three-arm intersection before reconstruction (T_16); (b) a roundabout after reconstruction (R_17); (c) a roundabout after five years of use (R_22). The models captured the change in the noise load in the outdoor environment around different intersection types.

CadnaA is a software program that is used for the calculation and assessment of noise and air pollution. The software calculates and predicts the noise impact in the vicinity of commercial and industrial sites, sports and leisure facilities, and traffic systems like roads and railways, airports and landing strips, and other noisy facilities. CadnaA is suitable for noise prediction in local studies as well as for detailed analyses of noise mapping scenarios in cities.

In the CadnaA program, it is possible to choose from several methods of calculating noise levels from different sources. For the modeling of noise propagation in the vicinity

of the proposed intersections, we used the "Nouvelle Méthode de Prévision du Bruit" (NMPB-1996).

The emission parameter used in the NMPB-1996 is the *A*-weighted sound power level per octave band $L_{Aw,i}$ of a point like sub-source $S_i$ in dB(A) [45]. This level is calculated from

$$L_{Aw,i} = 10\lg\left(10^{\frac{(E_{VL}+10\lg Q_{VL})}{10}} + 10^{\frac{(E_{PL}+10\lg Q_{PL})}{10}}\right) + 20dB + 10\lg l_i + R(i) \tag{4}$$

where $E_{VL}$ is the sound power level of the light vehicles in dB(A), $Q_{VL}$ is the traffic number of the light vehicles (max. mass m < 3500 kg) in veh/h, $E_{PL}$ is the sound power level of heavy vehicles in dB(A), $Q_{PL}$ is the number of heavy vehicles (max. mass m $\geq$ 3500 kg) in veh/h, $l_i$ is the length of the sub-source $S_i$ in m, $R(i)$ is the octave values of the reference spectrum for road noise in dB(A), and *i* is the running number of octaves.

In CadnaA, the overall traffic flow $Q$ (in vehicles/hour) and the percentage of heavy vehicles $p$% are specified. The relationships between those figures and the ones described above are

$$Q = Q_{VL} + Q_{PL} \tag{5}$$

$$p\% = \frac{Q_{PL}}{Q} \tag{6}$$

The calculated emission parameter is the *A*-weighted total sound power level per unit length $L'_{Aw,i}$ in dB(A), which is determined according to the modified equation:

$$L'_{Aw,i} = 10\lg\left(10^{\frac{(E_{VL}+10\lg Q_{VL})}{10}} + 10^{\frac{(E_{PL}+10\lg Q_{PL})}{10}}\right) + 20dB + \Psi \ [dB(A)] \tag{7}$$

This value includes the corrections for the vehicle type, traffic flow type, road gradient, and road surface $\Psi$ (see below).

The respective data for the hourly traffic $Q$ (veh/h) and the truck proportion $p$ (%) are given in the next table (Table 3). MDTD and the truck proportion $p$ data were taken from real measurements of the traffic volume attained during the noise measurement. The truck proportion $p$ (%) was used for all three modeled scenarios (T_16, R_17 and R_22) (Table 3), and was evaluated from real measurements of traffic flow.

**Table 3.** Hourly traffic volume of vehicles and percentages of trucks for various time periods used in the model of intersections for Žilina City.

| Road Type | Day (6–18 h) | | Evening (18–22 h) | | Night (22–6 h) | |
|---|---|---|---|---|---|---|
| | Q (veh/h) * | p (%) | Q (veh/h) * | p (%) | Q (veh/h) * | p (%) |
| Local Road | 0.062 * MDTD | 5 | 0.042 * MDTD | 3 | 0.011 * MDTD | 6 |

* According to VBUS [53].

The traffic volume determined by direct measurements with the radar counting device was 15,687 vehicles/24 h passing through the three-arm intersection (T_16) and 16,520 vehicles/24 h passing through the roundabout (R_17) on the selected working day (Table 4). Traffic volume was monitored during the noise measurement day. These data on the traffic volume and its direction were used for the noise load model in the vicinity of the intersection. The traffic volume measured at the roundabout in 2017 was also used for the model of the roundabout (R_22) in 2022.

Traffic routing through intersections and defining traffic volume for individual homogeneous sections is based on the traffic routing matrix obtained from the traffic routing survey (Table 5).

**Table 4.** Daily traffic volume (MDTD) at the entrances to the intersection detected during the working day at the three-arm intersection (T_16) and roundabout (R_17) in Žilina City.

| | Traffic Volume at the Intersection Entrance (veh/24 h) | | |
| --- | --- | --- | --- |
| Intersection/Entrance | A | B | C |
| T_16 | 5702 | 6763 | 3222 |
| R_17 | 5333 | 7311 | 3876 |

**Table 5.** Traffic routing matrix through the intersections (three-arm intersection, roundabout) in Žilina City.

| Three-Arm Intersection (T_16) | | | |
| --- | --- | --- | --- |
| Entrance to the Intersection | A | B | C |
| A | 0% | 69% | 31% |
| B | 57% | 0% | 43% |
| C | 37% | 63% | 0% |
| Roundabout (R_17, R_22) | | | |
| Entrance to the Intersection | A | B | C |
| A | 0% | 69% | 31% |
| B | 63% | 0% | 37% |
| C | 43% | 57% | 0% |

With NMPB, the correction for different road surfaces is already included in the calculation of the emission level. The influences of different types of pavement surfaces and the effects of their surface wear on noise levels have been topics of significant research by the authors [9,14,54]. Different properties of asphalt concrete (AC) pavement surface were used for each scenario T_16, R_17, and R_22. AC-old was used for T_16, AC-new for R_17, and AC-partially smoothed for R_22. Based on the conducted research, corrections (Sound reduction indices) for the new road surface (R_17) of 3 dB, and 2 dB for the road surface after five years (R_22) were used.

The speed of the traffic flow (cars and trucks) was chosen according to the measurements of traffic volume at the individual entrances to the intersection (Table 6). The overall average speed of vehicles passing through the three-arm intersection was higher than that at the roundabout. The average vehicle speeds on the arms of the roundabout were used for models R_17 and R_22.

**Table 6.** Average driving speeds of cars and trucks on the intersection arms (profiles A, B, and C) of intersections in Žilina city.

| Intersection Arms (Profiles) | The Average Speed of Vehicles [km/h] | | | |
| --- | --- | --- | --- | --- |
| | Three-Arm Intersection (T_16) | | Roundabout (R_17, R_22) | |
| | Passenger Cars | Freight Vehicles | Passenger Cars | Freight Vehicles |
| B | 52 | 48 | 50 | 43 |
| A | 44 | 37 | 40 | 32 |
| C | 43 | 33 | 35 | 25 |

The wind rose values obtained from real measurements of wind speed and direction were used to model noise propagation in the vicinity of the aforementioned intersections (Figure 5). At the three-arm intersection, the prevailing winds were from the south and southeast directions. At the roundabout, the prevailing winds were from the southwest and east directions.

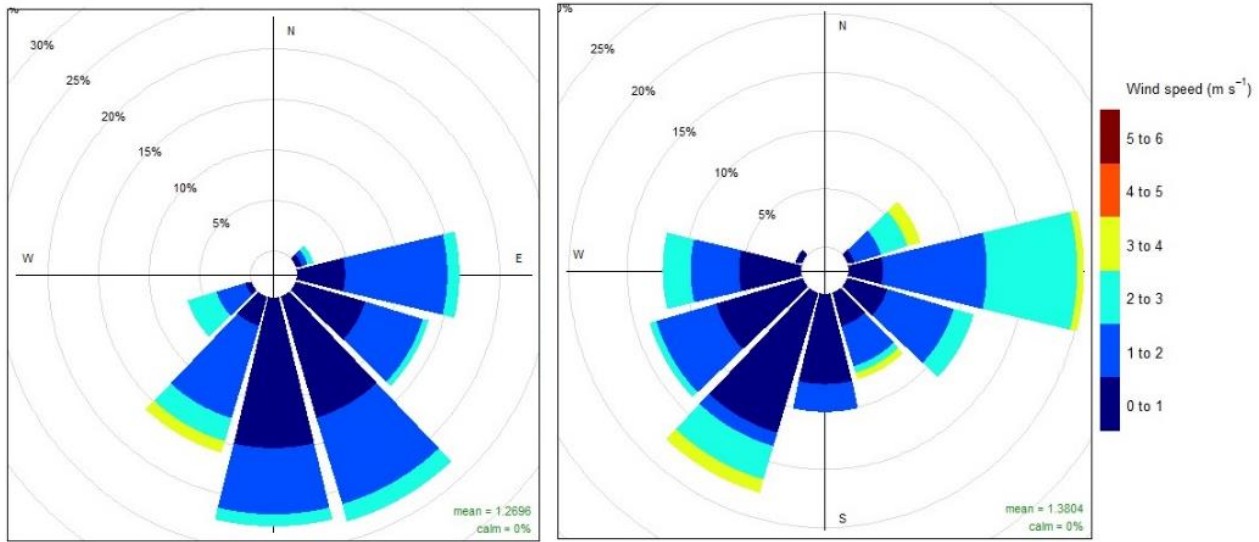

**Figure 5.** Wind rose values for the prevailing winds at the three-arm intersection (**left**) and round-about (**right**).

## 3. Results

### 3.1. Field Measurements

The measurements of the equivalent noise levels conducted before and after the reconstruction of intersections into roundabouts in Michalovce city are shown in Table 7 and Figure 6.

**Table 7.** $L_{Aeq}$ from traffic at the intersection of R1, R2 and R3 before and after the reconstruction.

| Equivalent Noise Level from Road Traffic [dB] at the Intersection before and after Reconstruction to R1 | | | | |
|---|---|---|---|---|
| **Before** | $\mathbf{L_{Aeq,15\ min}}$ | | **After** | $\mathbf{L_{Aeq,15\ min}}$ | |
| 29 May 2006 $11^{00}$ to $12^{00}$ | 59.5 61.1 | 61.1 59.8 | 12 March 2007 $10^{30}$ to $11^{15}$ | 58.6 58.3 | 58.7 - |
| **29.5.2006 $L_{Aeq,1\ h}$ = 60.4 dB and 12.3.2007 $L_{Aeq,45\ min}$ = 58.5 dB** | | | | |
| Equivalent Noise Level from Road Traffic [dB] at the Intersection before and after Reconstruction to R2 | | | | |
| 29 May 2006 $12^{30}$ to $13^{30}$ | 63.7 60.0 | 60.0 61.0 | 12 March 2007 $12^{00}$ to $13^{00}$ | 58.9 59.5 | 58.5 - |
| **29.5.2006 $L_{Aeq,1\ h}$ = 61.5 dB and 12.3.2007 $L_{Aeq,45\ min}$ = 59.0 dB** | | | | |
| Equivalent Noise Level from Road Traffic [dB] at the Intersection before and after Reconstruction to R3 | | | | |
| 12 March 2007 $12^{00}$ to $13^{00}$ | 56.6 58.3 | 57.4 57.9 | 29 April 2008 $10^{30}$ to $11^{15}$ | 55.3 54.5 | 55.0 54.4 |
| **12.3.2007 $L_{Aeq,1\ h}$ = 57.6 dB and 29.4.2008 $L_{Aeq,45\ min}$ = 54.8 dB** | | | | |

Before rehabilitation, the reconstructed intersections (R1, R2, and R3) in Michalovce had a good quality wearing surface course without repairs, dents, or cracks. The conducted noise pollution measurements found that the implementation of roundabouts R1 and R2 reduced the noise level by an average of 2.2 dB. One to three-octave noise analyses were not available for R3, and hourly measurements of equivalent noise levels found a reduction of 2.8 dB. This high value compared to those of R1 and R2 was caused by the poor-quality road surface of the intersection pavements before their reconstruction into a roundabout. Additionally, according to statements of local citizens living near R3, there was a significant reduction in noise pollution in the vicinity of the roundabout.

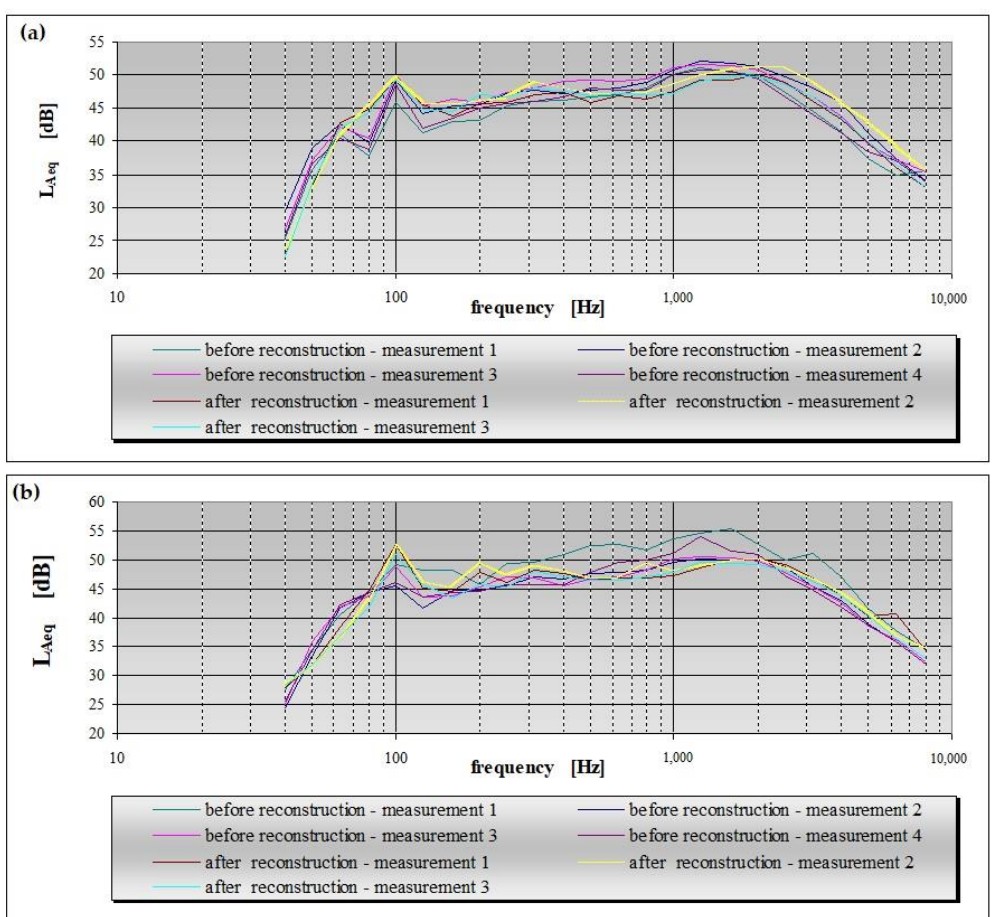

**Figure 6.** Comparison in the 1–3 octave bands of equivalent noise level $L_{Aeq}$ from road traffic before and after the reconstruction of intersections in Michalovce: (**a**) R1 and (**b**) R2.

Measurements of the noise load in the city of Žilina revealed a decrease in noise levels after the reconstruction of the three-arm intersection into a roundabout. Reductions in the noise level were by 3.4 dB at monitoring station MS1 and 2.9 dB at MS2. Measurements after five years of use of the roundabout revealed a slight increase in the noise load in the vicinity of the intersection. Compared to the original three-arm intersection, the decreases were 2.6 dB at MS1 and 2.1 dB at MS2 (Table 8, Figure 7).

**Table 8.** Equivalent noise levels at the assessed points obtained from modeling $L_{Aeq,day}$, $L_{Aeq,evening}$, and $L_{Aeq,night}$ and from real measurements $L_{Aeq,day,m}$ conducted in Žilina city.

| Intersection (Scenario) | Modelling Point/ Measuring Station | $L_{Aeq,day}$ [dB] | $L_{Aeq,evening}$ [dB] | $L_{Aeq,night}$ [dB] | $L_{Aeq,day,m}$ [dB] |
|---|---|---|---|---|---|
| | MP1 | 62.9 | 60.6 | 56.9 | - |
| T_16 | MS1 | 70.7 | 67.9 | 63.8 | 68.9 |
| | MS2 | 69.0 | 66.1 | 62.3 | 68.0 |
| | MP1 | 60.0 | 57.5 | 54.2 | - |
| R_17 | MS1 | 67.4 | 64.3 | 60.7 | 65.5 |
| | MS2 | 67.5 | 64.4 | 60.8 | 65.1 |
| | MP1 | 61.0 | 58.5 | 55.2 | - |
| R_22 | MS1 | 68.4 | 65.3 | 61.7 | 66.3 |
| | MS2 | 68.5 | 65.4 | 61.8 | 65.9 |

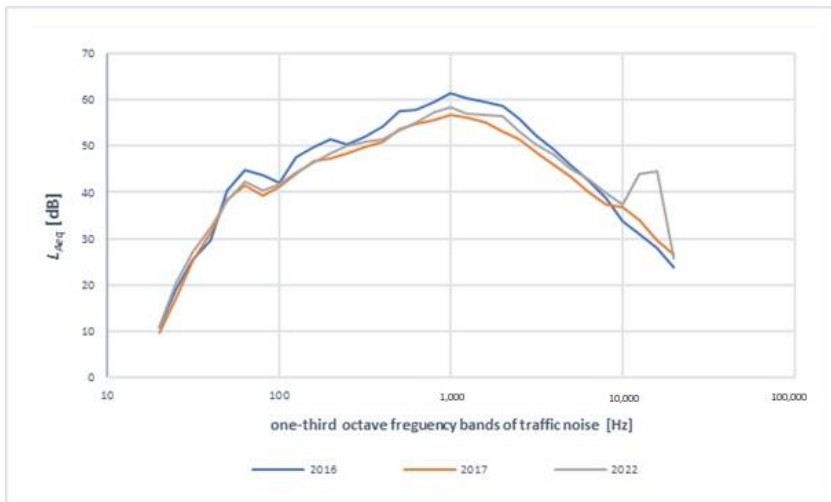

**Figure 7.** Comparison in 1–3 octave bands for equivalent noise levels from road traffic before and after the reconstruction of an intersection in Zilina at MS1 for years 2016 (before reconstruction), 2017 (immediately after reconstruction), and 2022 (five years after reconstruction).

### 3.2. Road Traffic Noise Modelling

The 3D map based on the "OpenStreetMap contributors" source was used as the basis for the model. This included the basic geometric structures of the intersections, surrounding buildings, greenery, and other types of infrastructure. The map background was transformed into the appropriate coordination system "Krovak S-JTSK (Ferro South/West positive) coordinate". Individual intersections were divided into homogeneous sections from the point of view of the traffic flow characteristics (traffic volume, speed, and density), road gradient, etc. (Figure 8, Table S1). Individual road sections were assigned the following parameters: traffic volume, road type (local road), traffic flow speed, road surface, traffic flow type, road gradient, and wind rose. These parameters affect the resulting noise levels around the road (according to the principles stated in Section 2.5).

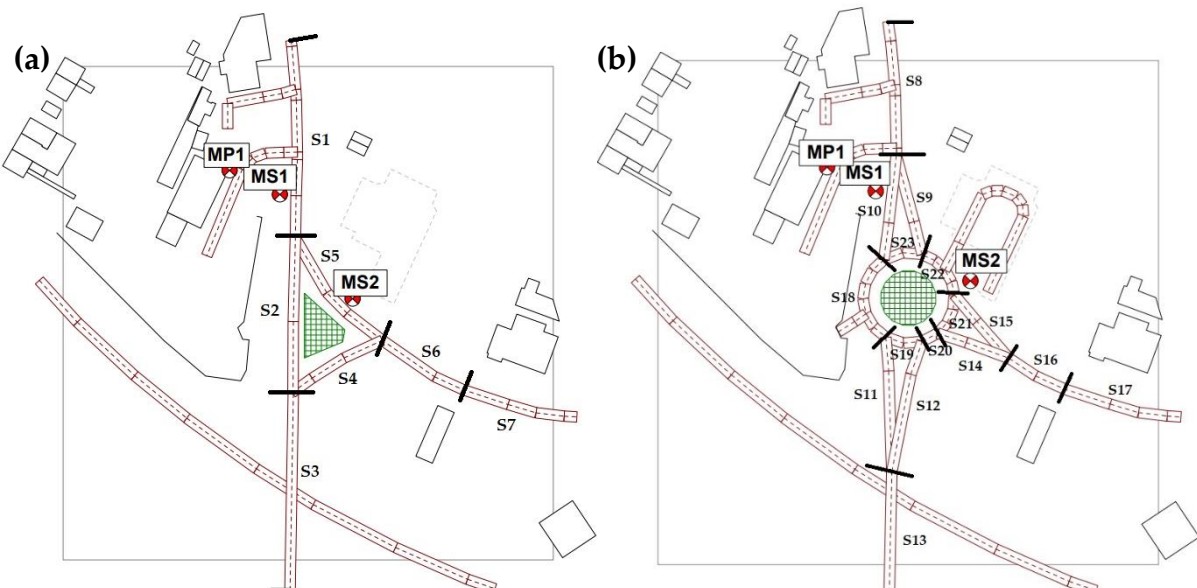

**Figure 8.** Distribution of homogeneous intersection sections (**a**) three-arm intersection-T_16, (**b**) roundabout-R_17, R_22) in the city of Žilina.

The parameters of the surrounding buildings and other parts of the infrastructure around the intersection were also defined: height, reflectivity parameter, or noise absorption

of the surrounding areas. Calculations were conducted for the selected area of interest and also for the selected points (measuring stations and modeling points).

Two measuring stations (MS1 and MS2) were chosen as reference points for the assessment, and real noise load measurements were also carried out. At the same time, one reference modeling point (MP1) was chosen near the exposed building for the comparison of the noise load coming from road traffic at the proposed intersections (Figure 3).

Noise level limits for the daytime, evening, and night-time (Table 1) were defined for the selected points. Points marked in red represent areas exceeding the limit value for the noise level (Figures 9–11).

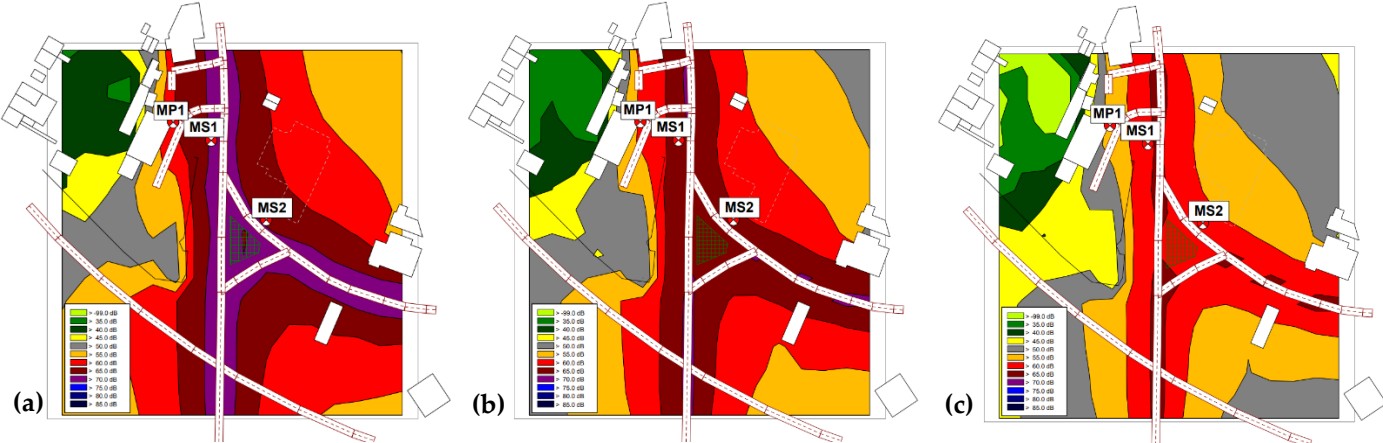

**Figure 9.** Model of the spreading of noise emissions from road traffic in the vicinity of the original three-arm intersection in the city of Žilina (T_16) during the (**a**) daytime, (**b**) evening and (**c**) night-time.

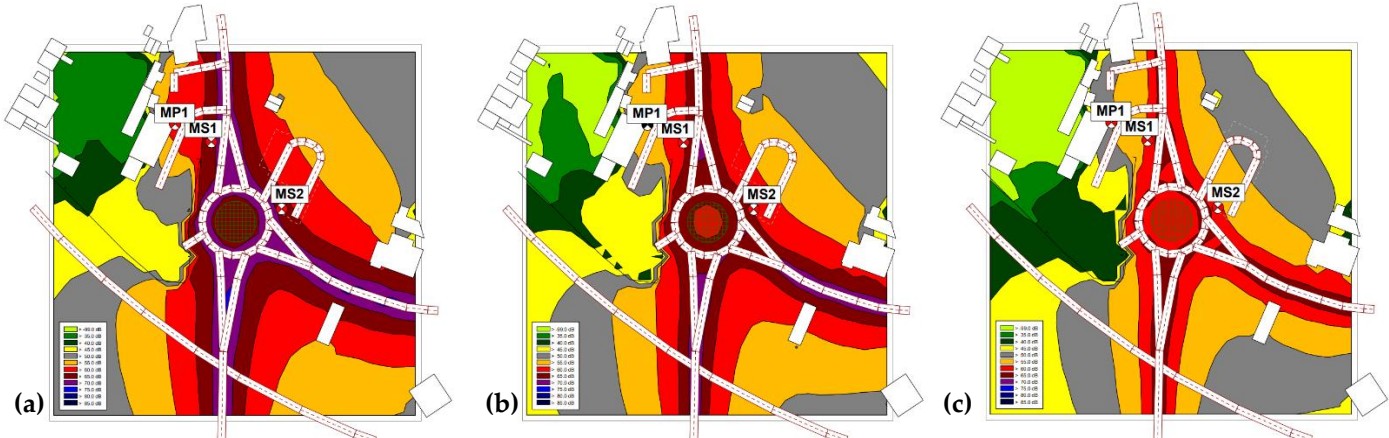

**Figure 10.** Model of the spreading of noise emissions from road traffic in the vicinity of the new roundabout in the city of Žilina (R_17) during the (**a**) daytime, (**b**) evening and (**c**) night-time.

Modeling of the noise load in the vicinity of the three-arm and roundabout intersection revealed differences in noise propagation from road traffic in the outdoor environment. Differences in the generation of noise from road traffic according to the design of the intersection were found. These may have an impact on the surrounding development of family or apartment buildings.

The equivalent noise level was calculated using the model for three reference parts of the day: daytime $L_{Aeq,day}$, evening $L_{Aeq,evening}$, and night-time $L_{Aeq,night}$. At all assessed points and for all assessed parts of the day (Table 8) limit values for the outdoor environment were exceeded (Table 1).

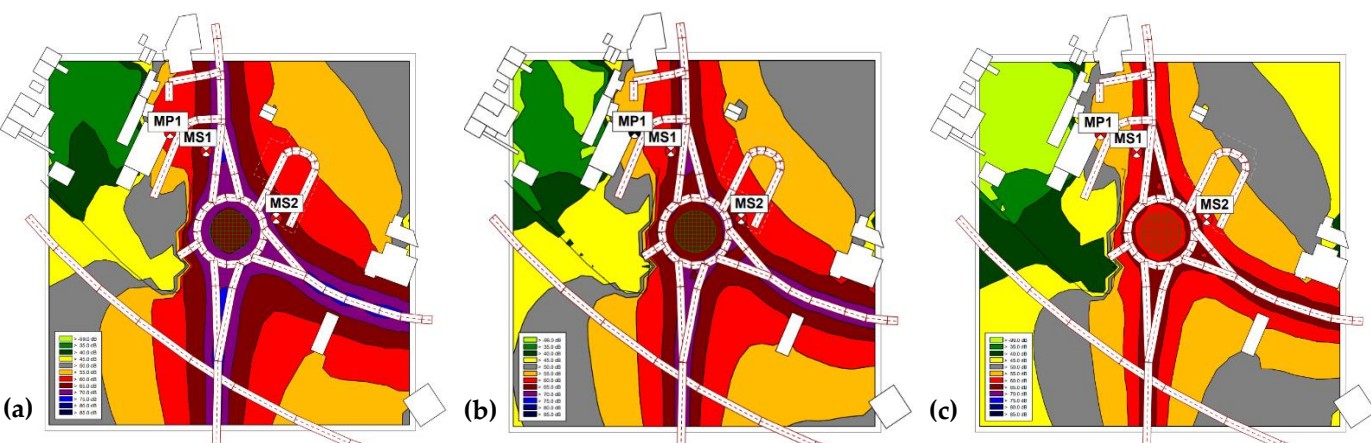

**Figure 11.** Model of the spreading of noise emissions from road traffic in the vicinity of the roundabout after 5 years of use in the city of Žilina (R_22) during the (**a**) daytime, (**b**) evening, and (**c**) night-time.

The models showed a reduction in the noise load at the roundabout compared to the three-arm intersection at the assessed locations MP1, MS1, and MS2 (Figures 9–11). The noise level reduction for the daytime ($L_{Aeq,day}$) for MS1 (R_17) compared with MS1 (T_16) was 3.3 dB; for MS2 (R_17) compared with MS2 (T_16), it was 1.5 dB; and for MP1 (R_17) compared with MP1 (T_16), it was 2.9 dB (Table 8, Figures 9 and 10). After five years of roundabout use, we observed a slight increase in the noise level at the assessed locations. For all stations, this increase was 1 dB. This may have been due to wear and smoothing of the road surface [50]. Similar results were also obtained by direct field measurements at sites MS1 and MS2 (Table 8). The models showed lower exposure of the given area to noise emissions, which was observed by layering the isolines of the noise level in the vicinity of intersections (Figures 9 and 10).

In the case of the new roundabout in the city of Žilina, we observed, according to the model, that the limit value of the noise level (Table 1) was not exceeded in the evening at station MP1 (Table 8, Figures 10b and 11b). This station was chosen as a reference modeling point for assessing the exposure of a nearby residential building to noise from road traffic. The noise levels reached, according to the model, were 57.5 dB for R_17 and 58.5 dB for R_22 (Table 8).

## 4. Discussion

When creating this article, the authors used their long-term research activities in the field of road traffic noise measurement and prediction. As part of the research, close attention was paid to low-noise road surfaces [55,56], the influence of the asphalt mixture composition [57], the condition of the surface of asphalt roads [9,25], and the effect of intersections [58,59] on the traffic noise level. More than 20 years of research activity have contributed to the improved precision of noise predictions for Central Europe conditions. This ensures the compatibility of the most commonly used prediction methodologies [60–62].

In this paper, we presented methods of traffic noise reduction around urban intersections (UI). In the case of UI, the possibility for reduction through road traffic noise reduction devices [63] and low-noise asphalt covers [55,56,64] is very limited. Noise barriers are very rarely used in urban areas, due to such factors as the lack of space, traffic safety, architectural, and esthetical factors. The use of low-noise asphalt pavements is limited in urban areas primarily due to the high cost of production and maintenance of these covers, their limited durability and lower efficiency compared to roads outside urban areas. For this reason, it was necessary to look for alternative solutions for reducing the noise load of the external environment surrounding residential buildings through a contextual intersection design. For the above reasons, from the aspect of traffic sustainability, it makes sense to use roundabouts in the vicinity of residential buildings. A possible traffic noise reduction of 2–6 dB due to the reconstruction of intersections to roundabouts [58,59], depending on

the condition of the wearing course layers of the reconstructed intersections was clearly demonstrated. Other authors have presented similar findings using the micro-simulation model. Under saturated conditions, there was a noise reduction of 2.5 dB at roundabouts compared with signalized intersections [65,66]. The average reduction induced by the roundabouts reached 2.7 dB with a maximum reduction of 5 dB for the signalized crossing. The average reduction was 1.8 dB. The resulting mean traffic noise levels during the weekday and weekends at the signalized intersection exceeded those of the roundabout by 3.5 and 3.7 dB(A), respectively. The results also indicate that the presence of heavy vehicles can significantly impact the observed traffic noise level, despite accounting for a small fraction of the traffic composition [66–68].

In the paper [69], it is stated that roundabouts are acknowledged as a beneficial countermeasure which has contributed to the improvement of traffic safety in Central European countries. A comparative study done on the sample of roundabouts in four Middle-European countries (Czech Republic, Hungary, Poland, and Slovakia) was based on the analyses of accidents, traffic, and geometry data. The injury accident frequency is positively associated with the effect of traffic volume and apron width, while negatively associated with deflection in terms of both entry and deviation angles [69,70]. Single-lane roundabouts are particularly popular due to their ability to increase traffic flow and improvement of safety. From the aspect of the contextual design of intersections, the roundabout central island gives such intersections an aesthetic advantage (display of local arts, plant and plant-rock compositions, etc.) over the traditional ones [71]. It is largely accepted that converting a signalised or non-signalised intersection into a roundabout will decrease the noise level [35]. According to Croatian authors [38], the roundabouts have proven to be very successful at improving safety (due to the reduced number of conflict points and lower speed compared to the traditional intersections) while enhancing mobility by reducing the total delay compared to other controlled intersections. The introduction of modern roundabouts in road networks has proved to be a good solution for reaching transportation sustainability goals. However, one should be aware that only a well-designed roundabout, considerate of a larger number of passing vehicles, a more significant percentage of heavy vehicles, and higher driving speed, will be effective as a noise abatement measure [38].

Generally, the principal environmental advantages of roundabouts in the vicinity of residential buildings include a higher safety level, consequent reductions in acoustic and atmospheric pollution, contextual adaptation to the urban environment, and ease of insertion into urban sites, where many squares are already configured in a ring scheme [37,38,58,65,72–74].

## 5. Conclusions

The authors draw attention to the fact that the reduction of traffic noise induced by the reconstruction of intersections into roundabouts consists of several factors. Both the new road surface of the intersection [15,57] and the change in traffic processes (traffic volume, speed of the traffic flow, and composition of the traffic flow) [27,35,38,72] contribute to the presented reduction in traffic noise at the roundabout, as determined using direct measurements and modelling. In the case of the presented intersections, minimal differences in the traffic volume and the composition of the traffic flow (proportion of trucks) were recorded before and after the reconstruction. The authors showed that a gradual increase in noise emissions was induced by the degradation of the asphalt surface morphology: on average, 1 dB after a million vehicles had passed, 3 dB after 10 million, and 5 dB after 100 million [14]. Within this paper, the authors presented their own research results, which show a reduction in traffic noise caused by the reconstruction of a non-signalized intersection into a roundabout. In the case of intersections in the city of Michalovce, decreases in the noise level detected by field measurements at the new roundabout were 1.9 dB for R1, 2.5 dB for R2, and 2.8 dB for R3. In the case of measurements in the city of Žilina, field measurements at the new roundabout detected decreases in the noise level of 3.4 dB at

MS1 and 2.9 dB at MS2. Noise dispersion in the vicinity of intersections in the city of Žilina was also modelled using the CadnaA program. Decreases in the noise level at the new roundabout were 3.3 dB at MS1, 1.5 dB at MS2, and 2.9 dB at MP1. After five years of use of the Zilina roundabout, an average increase of 1 dB occurred due to wear and smoothing of the asphalt pavement surface. The authors assume that the presented research results will contribute to the improvement of the contextual solution for urban intersections under Slovak conditions by integrating acoustic, psychoacoustic, architectural, environmental, and economic aspects, providing a holistic approach to road design.

**Supplementary Materials:** The following supporting information can be downloaded at: https://www.mdpi.com/article/10.3390/app12178878/s1, Table S1: Parameters of homogenous sections (road segments) used in road traffic noise models.

**Author Contributions:** Conceptualization, D.J. and M.D.; Data curation, K.H., P.P. and D.B.; Formal analysis, M.D.; Investigation, D.J., M.D., K.H., P.P. and D.B.; Methodology, D.J. and M.D.; Resources, K.H., P.P. and D.B.; Software, D.J.; Supervision, M.D.; Validation, D.J. and M.D.; Visualization, D.J. and M.D.; Writing—original draft, M.D.; and Writing—review & editing, D.J. and M.D. All authors have read and agreed to the published version of the manuscript.

**Funding:** This research received no external funding.

**Institutional Review Board Statement:** Not applicable.

**Informed Consent Statement:** Not applicable.

**Acknowledgments:** The paper was supported by a VEGA 1/0337/22 grant entitled "Analysis of the influence of the pavement surface texture on skid friction, road safety, and the potential of road dust resuspension".

**Conflicts of Interest:** The authors declare no conflict of interest.

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
