# Peer review of "Influence of the Urban Intersection Reconstruction on the Reduction of Road Traffic Noise Pollution"

_applsci, doi:10.3390/app12178878_

Round 1

Reviewer 1 Report

(1) It seems that main data are referred to the published papers, such as data in Figures 1 and 2, Tables 1 and 2. The authors conduct some new experiments at Figures 3 and 4. The authors should seriously describe the meaning of their work, since many similar works had already been conducted, and the achieved conclusions are similar with the previous research.

(2) The authors should explain the meaning of T_16, R_17, R-22, and other abbreviations in the text, otherwise the presentations are difficult to understand.

(3) The major achievement is obtained by the simulation software CadnaA, so please give more details about the simulation parameters and conditions.

(4) The conclusions are too long. Please simplify the major achievements obtained in this manuscript.

(5) There are too many spelling, typing and grammar mistakes in the text. Please check the whole manuscript carefully.

(6) The marks of some references are inconsistent. For example, the reference marked at the Table 2 is [47], but in the text the relevant contents are marked by [45] in the paragraph above. Please check all the references carefully.

(7) For the references in the introduction section, the author should describe the main achievements obtained by previous researches, instead of just give a simple mark.

Author Response

Dear reviewer,
thank you for your feedback and comments. I hope we understood all your comments correctly. We tried our best to incorporate all suggestions and recommendations into the article. A more detailed response to your review is in the attached file.
Thank you and I wish you all the best.

Reviewer 2 Report

In my opinion, the authors failed to adequately present the topic of the paper, which they gave in the Introduction: „The paper present in Slovak conditions of a new approach to the design of UC, which integrates acoustic, psychoacoustic, architectural, environmental, and economic aspects of a holistic approach [25-27] to the design of these engineering structures (CD of UC ), similar to what is in the crossroads contextual design of the Polish 58 [28-30], Czech [31, 32], and other authors [33-38].“. In this paper, authors merely presented (rather poorly and without consistency) the results of their research on road traffic noise reduction potential of urban roundabouts, i.e., the impact that the reconstruction of 4 standard (signalized?) intersections to roundabouts had on the reduction of noise levels in the vicinity of these intersections. According to the authors, this impact was both measured and modelled. Currently, the paper is only at a draft level, as the input data and the results for all analyzed intersections are either not presented clearly, or not presented at all. Also, the aforementioned holistic approach to the design of urban crossroads is not presented. Therefore, my recommendation is to reject the paper.

Author Response

Dear reviewer,
thank you for your feedback and comments. The article was modified based on your comments and the comments of other reviewers. We tried our best to incorporate all suggestions and recommendations into the article. I hope that the reaction to your review will be sufficiently convincing about the validity of this study. A more detailed response to your review is in the attached file.
Thank you and I wish you all the best.

Reviewer 3 Report

1) Improve in the title should be changed to reduce, or else it is totally the opposite meaning.

2) The logic of the abstract should be significantly improved. As a normal research paper, the information should appear in the order of research background, research aim, research method, major results, significance.

3) Line 31-32, do you have any reference for this statement, as sometimes air pollution bothers residents more.

4) The illustration of some technical process should be simple and clear. Very basic knowledge is not necessary to provide, such as the illustration about LAeq.

5) Conclusions should be more concise, and some discussion should be separated as a independent part. The last section could be Discussion and Conclusions.

6) The language should be improved. Grammatical problems exited in many places, such as Line 43, paid to...; Line 49 ..., which is a....; Line 60, 64, 368, ect...

Author Response

(The authors gave the same response as above.)

Round 2

Reviewer 1 Report

Please highlight in yellow or change color of the words for the corrected part in the revised manuscript. Otherwise, it is difficult for the reviewers to check whether you have revised the manuscript and which parts have been corrected. This is also the normal requirement in this Journal for the revised manuscript.

Before you have marked the corresponding corrections in the revised manuscript relative to the original manuscript, it is difficult for me to check whether your revision is satisfactory.

Author Response

Dear reviewer. There must have been some misunderstanding. Of course, I followed the instructions of the journal and the editor. I used the "Track changes" function ("Any revisions to the manuscript should be marked up using the “Track Changes” function if you are using MS Word/LaTeX, such that any changes can be easily viewed by the editors and reviewers") in the corrected manuscript and uploaded the modified manuscript to the system. I hope it will be ok now.

Reviewer 2 Report

My recommendation is to reconsider the paper after the following issues are addressed sufficiently.

GENERAL REMARKS

Abstract, Lines 1113

It is true that noise barriers are rarely used in urban areas. This is due to the lack of space for their installation, traffic safety (intersection visibility requirements), architectural factors, as well as the fact that they represent a barrier for pedestrians and cyclists. On the other hand, the use of low-noise pavement surfaces is limited in urban areas primarily due to the financial factors (high cost of production, installation, and maintenance of these types of pavements), their limited durability in a colder climate, and the fact that they are more efficient as a noise-reduction measure on roads with vehicle speed over 50 km/h (road outside urban areas).

I believe that the authors should rewrite the Abstract and Conclusions to address the abovementioned issues.

Chapter 2.5. Modeling noise emissions

The authors used software CadnaA and analytic noise model NMPB-1996 for road traffic noise modeling. Unfortunately, they did not describe the model with sufficient detail to allow others to replicate and build on published results. Namely, they failed to mention the position of noise sources and their homogeneous subsections. This data is very important as analytic noise models attempt to capture the impact of interrupted traffic on the average vehicle speed profile by splitting each road section into subsections where vehicles are assumed to have a constant average speed and homogeneous traffic flow conditions (for further details see your reference [42]). Furthermore, it is unclear for what scenario of the roundabout the average speed of vehicles on the approaches was provided.

As for the heavy vehicles proportion definition, the authors used the German static model RLS-90. The main issues with this are as follows.

(1) RLS-90 model is nowadays obsolete and replaced with the RLS-19.

(2) In RLS-90 heavy vehicles are the ones with a total weight of over 2,8 t (RLS-19, NMPB-1996: heavy vehicles are the ones with a total weight of over 3,5 t).

(3) RLS-90 defines only the period “day” and “night”, so it is unclear where the data for the period evening comes from.

(4) If the traffic counts were conducted during measurements, this data should be used in modeling (see “Good practice guide for strategic noise mapping”).

I believe that the authors should rewrite this chapter to address/resolve the abovementioned issues.

SPECIFIC REMARKS

Line 12: For clarity’s sake, replace “pavement cover” with “Pavement Surface”, “Surface Course”, “Surface Layer”, “Wearing Course”, or “Weathering Course” (this replacement should be conducted in the entire paper).

Line 15: Please add “unsignalized” before “urban intersections”.

Lines 3334: Since the topic of the paper is road traffic noise pollution, I believe that this sentence, including references 710, is a surplus (also, inappropriate self-citations by authors are detected).

Line 46: Please use “low-noise”, as you did in the rest of the paper.

Line 58: Please note that “rehabilitation” is not a synonym for “reconstruction” in civil engineering. Rehabilitation means “to restore to near original condition” (for example, by introducing the asphalt overlay or asphalt replacement). Reconstruction means “to comprehensively rebuild to a new condition with current criteria” (for example, by replacing a conventional intersection with a roundabout).

Line 73: Typo, “… a previous study [31], …”.

Lines 7885: For clarity’s sake, please replace “non-light-controlled” with “unsignalized”, and “traffic light intersections” with “signalized intersections” (see also lines 106, 118).

Lines 8990: “The reconstruction of a signalized intersection to a roundabout…”.

Lines 9093: Please replace “slower speed” with “lower speed”. Reduction and increase in noise levels should be listed as absolute values (i.e., without the signs “-“ and “+”).

Line 142: Add definitions for p and p0 (“where p is …, and p0 is …”) after the equation (1).

Line 147: Add a definition for T after the equation (2).

Line 156: I would recommend changing the title to “Chapter 2.2. Admissible road traffic noise levels in the Slovak Republic”, and focusing this chapter on road noise, i.e., excluding data for the other noise sources from the paper.

Line 171: What does “LAeq,a” stands for in Table 1?

Line 183: The topic of this paper is outdoor (road traffic) noise sources, so I would suggest excluding the permissible noise levels from indoor sources from Table 2 and arranging the data only in the text body (i.e., remove the table 2). Why there are no permissible values for noise from the outdoor environment for the indoor area B category? What does “LAeq,p” stands for?

Line 193: Data in Figure 2 is obsolete (from 2012). Please provide current data for road traffic noise and all EU member states or exclude this figure from the paper.

Line 194: Please replace “transverse intersections” with “standard intersections” or “conventional intersections”.

Line 212: What was the wearing course on the first and second intersection in Michalovce before the reconstruction?

Line 214: What was the wearing course on the third intersection in Michalovce (before and after the reconstruction)?

Lines 215232: Please provide information on the time and duration of all three noise measurements (at least the time of day) and whether the data on traffic (load, composition, speed) was collected simultaneously with noise measurements.

Line 255: Please remove “emissions” from the title, because the last paragraph in this chapter (Lines 329328) refers to propagation modeling.

Line 257: It is advisable to emphasize at this point that the noise model was created for an intersection in Žilina (singular, not plural!) and for 3 scenarios.

Lines 303305: According to lines 300303, data on the truck proportion (p) was taken from the German standard RLS-90. Which statement is true?

Line 306: In the German standard RLS-90 (as well as in the new standard RLS-19) only two periods for 24h-day are given - period “day” (6-22) and period “night” (22-6). Where does your data for “evening” come from?

Lines 309311: When was the traffic load given in this sentence measured? What about different scenarios (was the same traffic load used for scenarios R_17 and R_22)?

Lines 314316: Data in Figure 6 should be arranged in the table (only numbers).

Line 326: Table 4 - was the same average speed used for scenarios R_17 and R_22?

Line 344: Table 5 – where is the data for R3? Is the time of measurement for R2 on 12. 3. 2007. correct (the same as the time for measurement on R1)?

Lines 346354: Were these intersections rehabilitated or reconstructed? What was the “high-quality wearing course”? Please describe all pavement surfaces.

Line 355: Figure 8 – data for R3 is missing. Leave only the diagram for LA,eq, “The decisive characteristic … of sound (noise)”.

Line 356: Figure 9 – leave only the diagram for LA,eq.

Line 400: Table 6 – are these values equivalent, A-weighted noise levels?

Lines 407411: The increase in noise levels could be caused by differences in noise source description - a larger number of passing vehicles, a more significant percentage of heavy vehicles, and higher driving speed (drivers are now used to this roundabout and are negotiating it at higher speed). Please elaborate on that (and address this issue in the Conclusions also).

Author Response

Dear reviewer. Thank you for the valuable and very helpful comments and recommendations, which undoubtedly raised the quality of the manuscript. I hope that we have responded to everything sufficiently and satisfactorily. Detailed responses to comments are in the attached MS word.

Round 3

Reviewer 1 Report

As my concerned problems and questions have been responded, and the corresponding corrections had been conducted in the revised manuscript, I suppose that this manuscript can be accepted.

Author Response

Dear Reviewer,
thank you for your comment and recommendation to accept this manuscript.

Reviewer 2 Report

I believe that the current title “Contextual design of urban intersections to reduce noise pollution in the outdoor environment of residential buildings” is not applicable, because the paper merely mentions CD. Furthermore, the authors did not discuss the influence of noise on residents or the influence of residential buildings on noise levels (for example their influence on noise propagation path). I would suggest “Influence of the urban intersection reconstruction on the reduction of road traffic noise pollution”.

Authors should address the following issues in the subsection „Road-traffic-related noise emission modelling“.

(1)   The title should be changed to „Road traffic noise modelling“, as the noise propagation is also included in this part of the paper (otherwise, there would be no calculation results).

(2)   Please provide the data on the number of vehicles, percentage of heavy vehicles, and vehicle speed for each road segment.

(3)   If the same parameter values for noise source in the emission models and the same meteorological conditions in propagation models were used for scenarios R_17 and R_22, the resulting noise levels for both scenarios should be equal. Did you change the type of pavement in model R_22? If yes, please state this in the model description. If not, there could be an error in the models (maybe in the noise source definition) – check your input.

Author Response

Dear reviewer,
thanks for the recommendations. Detailed responses are presented in "Response_Reviewer 2_Round 3" and included in the manuscript.

Round 4

Reviewer 2 Report

Please check references 38 and 39 (style correction, wrong authors).

Author Response

Dear reviewer,
I have checked and corrected the references. Thanks for all your comments and recommendations.